# Trust-Region Saliency-Guided Local Search for Interpretable Sequence Design at Fixed Edit Budgets

## Abstract

Discrete sequence design under a fixed edit budget can match target model outputs, but often returns dispersed, multi-cluster edits that are hard to interpret. We present **SAGE-TRSwap**, a saliency-guided trust-region local search that optimizes the *same prediction loss* as a Ledidi-style relaxation+pruning baseline (Schreiber et al., 2021) while biasing proposals toward high-attribution regions and enabling budget-preserving SWAP refinements. Across **12** regulatory targets/tracks and **5** random starts per target (60 runs per budget), SAGE-TRSwap sharply reduces edit span and cluster count at all budgets (e.g., mean span $852 \rightarrow 100$ and clusters $6.0 \rightarrow 1.2$ at $B=40$) while maintaining or improving absolute error.

## 1 Introduction

Modern sequence-to-function predictors enable in silico design by providing differentiable surrogates of regulatory activity (Zhou & Troyanskaya, 2015; Kelley et al., 2016; Avsec et al., 2021b;a). Model-guided DNA sequence design seeks edits to a starting sequence $x_0$ that drive a predictor $f(x)$ toward a target output $y^\star$ under an edit budget $B$.

Ledidi-style approaches solve this via a differentiable relaxation plus discretization/pruning (Schreiber et al., 2021), but we observe a recurring failure mode: solutions with similar final error can have radically different *structure*. Some edits concentrate within a short locus (suggesting a motif-grammar hypothesis), while others disperse across distant loci (harder to rationalize and implement).

We ask: *Can we select more interpretable solutions at fixed edit budgets without degrading target matching?* We propose SAGE-TRSwap, a lightweight saliency-guided local search that preserves the original objective while biasing edits toward compact, attribution-consistent regions and enabling budget-preserving SWAP refinements.

## 2 Problem and metrics

We minimize prediction loss under a Hamming budget:

$$\min_{x \in \{A,C,G,T\}^L} \mathcal{L}(f(x), y^\star) \quad \text{s.t. } d_H(x, x_0) \leq B. \tag{1}$$

We report final absolute error $|f(x) - y^\star|$ and edits used. For interpretability, we measure **span** $\max(E) - \min(E)$ and **clusters** (merge edited positions whose gaps are $\leq 20$ bp). For attribution agreement, we compute an Integrated Gradients (IG) (Sundararajan et al., 2017) saliency map on the squared-error objective $(f(x) - y^\star)^2$ and quantify **saliency alignment** as the fraction of edits falling within the top-$X\%$ of the *original* IG map (alignment curve); we also report AP **average precision (AP)**, and use DeepSHAP (Shrikumar et al., 2016; Lundberg & Lee, 2017) for local sequence-logo visualizations in case studies.

**Algorithm 1** SAGE-TRSwap

1. Initialize $x \leftarrow x_0$, $E \leftarrow \emptyset$, best $\leftarrow x$.
2. While $|E| < B$: compute IG on $(f(x) - y^\star)^2$, set $\mathcal{T}$=TopK, score a small candidate set of edits (biased to $\mathcal{T}$), accept only stable improvements; keep best-so-far.
3. SWAP: while improving, replace $i \in E$ with low saliency by $j \notin E$ with high saliency (prefer $j \in \mathcal{T}$).
4. Return best.

## 3 METHODS

### 3.1 VANILLA BASELINE (LEDIDI-STYLE RELAXATION + PRUNING)

Following Ledidi's design pattern, we optimize a continuous relaxation of the discrete sequence using per-position base logits (softmax over {A,C,G,T}) to minimize squared error $\mathcal{L}(f(x), y^\star) = (f(x) - y^\star)^2$ plus an edit regularizer that encourages staying close to $x_0$. We sweep a log-spaced grid of regularization strengths (Ledidi-style $\lambda$ sweep), discretize by argmax, select the lowest-loss feasible candidate, and if needed enforce $d_H(x, x_0) \leq B$ by reverting the least-salient edits (Simonyan et al., 2013).

### 3.2 SAGE-TRSWAP

SAGE-TRSwap keeps the same loss and budget but changes the discrete search policy. At each iteration, we compute an IG (Sundararajan et al., 2017) saliency map on $(f(x) - y^\star)^2$, define a trust region $\mathcal{T}$ as TopK saliency positions (optionally window-expanded), and propose a small batch of single-base edits biased toward $\mathcal{T}$ and proximity to current edits. We accept only stable improvements (trust-region rule) while tracking best-so-far. Once $|E| = B$, we apply SWAP moves that replace a low-saliency edited position with a high-saliency unedited position, keeping $|E|$ fixed.

## 4 EXPERIMENTAL SETUP

We evaluate pretrained BPNet(-lite) Torch checkpoints for 12 regulatory targets/tracks downloaded from a Zenodo release used as Ledidi examples (Schreiber, 2025). Models are loaded in PyTorch (eval mode) and, when compatible, wrapped with `bpnetlite`'s `ControlWrapper`/`CountWrapper` to standardize the forward interface. Model outputs may be tensors or tuple/list outputs; we extract a tensor and define a scalar objective by selecting a fixed output index (target_index= 0) for all methods. For each target, we sample 5 random one-hot DNA sequences $x_0$ of length $L$=2114 (seeds 42–46), compute $y_0 = f(x_0)$, and set the desired target to $y^\star = y_0 + \Delta$ with $\Delta = 4.0$. We evaluate fixed edit budgets $B \in \{20, 40, 80\}$, yielding $12 \times 5 = 60$ runs per budget. Saliency maps use Integrated Gradients on the squared-error objective $(f(x) - y^\star)^2$ with a uniform baseline (all bases set to $0.25$) and 16 IG steps; span and clusters use a 20-bp gap threshold. All runs are executed on a single NVIDIA H100 GPU per target. Motif analyses in case studies use PFMs from the JASPAR database (e.g., MA0036.4) (Castro-Mondragon et al., 2022).

## 5 RESULTS

Across all TF×seed pairs ($12 \times 5 = 60$ runs per budget), SAGE-TRSwap improves compactness by a large margin at every budget while maintaining competitive target matching (Table 1). Compared to the Vanilla baseline, mean span decreases from $623.3 \rightarrow 68.6$ (B=20), $852.3 \rightarrow 100.1$ (B=40), and $1043.5 \rightarrow 123.0$ (B=80), while mean clusters decrease from $3.4 \rightarrow 1.2$, $6.0 \rightarrow 1.2$, and $9.2 \rightarrow 1.2$, respectively. Absolute error also improves on average ($2.33 \rightarrow 1.77$ at B=20; $1.26 \rightarrow 0.91$ at B=40; $0.81 \rightarrow 0.61$ at B=80). Edit–saliency agreement increases substantially, reaching $\approx 1.0$ across budgets, indicating that SAGE-TRSwap concentrates edits within the most attribution-salient regions of the original sequence. Interestingly, Ledidi-ST achieves strong error at higher budgets but remains substantially less compact (large spans and many clusters), highlighting that better target matching alone does not guarantee interpretable edit structure.

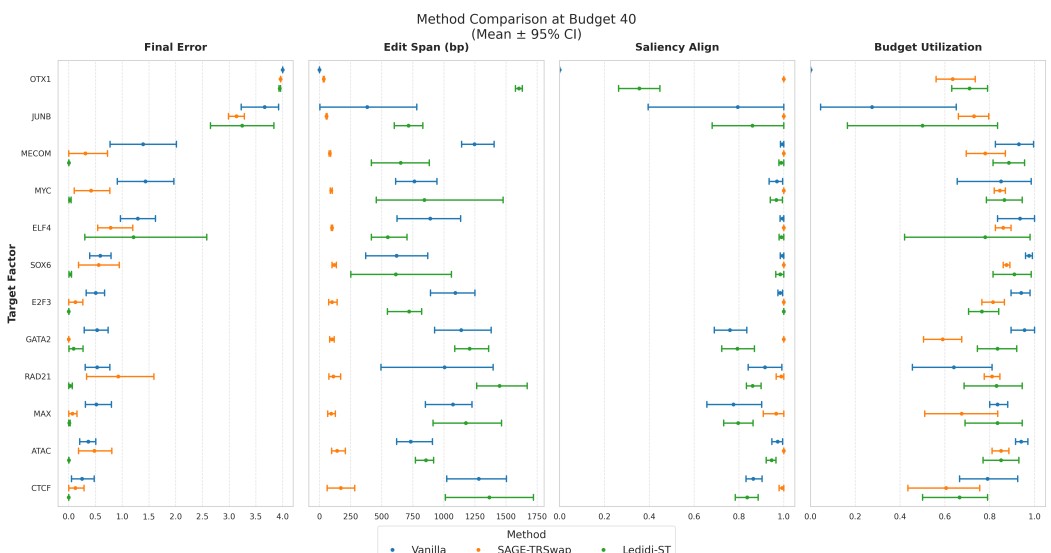

Figure 1: **Per-target performance and interpretability at fixed budget** $B = 40$ **(mean $\pm$ 95% CI over seeds).** For each target factor (row), points show the mean across seeds (42–46) and horizontal bars show 95% confidence intervals for three methods (Vanilla, SAGE-TRSwap, Ledidi-ST). Panels report (left to right): **Final Error** $|f(x) - y^\star|$ (lower is better), **Edit Span** (bp; lower is more localized), **Saliency Align** (fraction of edits that fall within the top-saliency region of the original IG map; higher is better), and **Budget Utilization** (fraction of the 40-edit budget used). Targets are ordered by the median Vanilla final error to align "harder" to "easier" targets visually.

| Method | Budget | Final_Error↓ | Span↓ | Clusters↓ | Final_Edits↓ | Budget_Util↓ | Saliency_Align↑ |
|---|---|---|---|---|---|---|---|
| Vanilla | 20 | $2.33 \pm 1.14$ | $623.3 \pm 488.6$ | $3.4 \pm 2.3$ | $13.8 \pm 6.5$ | $0.69 \pm 0.33$ | $0.88 \pm 0.30$ |
| SAGE-TRSwap | 20 | $\mathbf{1.77 \pm 1.14}$ | $\mathbf{68.6 \pm 58.2}$ | $\mathbf{1.2 \pm 0.4}$ | $16.8 \pm 1.6$ | $0.84 \pm 0.08$ | $\mathbf{1.00 \pm 0.00}$ |
| Ledidi-ST | 20 | $2.54 \pm 1.64$ | $583.3 \pm 508.4$ | $3.7 \pm 2.8$ | $\mathbf{10.6 \pm 8.5}$ | $\mathbf{0.53 \pm 0.43}$ | $0.87 \pm 0.22$ |
| Vanilla | 40 | $1.26 \pm 1.28$ | $852.3 \pm 465.2$ | $6.0 \pm 4.1$ | $\mathbf{30.2 \pm 13.4}$ | $\mathbf{0.76 \pm 0.33}$ | $0.84 \pm 0.30$ |
| SAGE-TRSwap | 40 | $0.91 \pm 1.28$ | $\mathbf{100.1 \pm 59.7}$ | $\mathbf{1.2 \pm 0.5}$ | $30.2 \pm 5.7$ | $0.76 \pm 0.14$ | $\mathbf{1.00 \pm 0.02}$ |
| Ledidi-ST | 40 | $\mathbf{0.71 \pm 1.42}$ | $979.3 \pm 463.3$ | $7.0 \pm 3.7$ | $31.4 \pm 8.7$ | $0.79 \pm 0.22$ | $0.86 \pm 0.19$ |
| Vanilla | 80 | $0.81 \pm 1.38$ | $1043.5 \pm 481.1$ | $9.2 \pm 6.0$ | $51.1 \pm 25.3$ | $0.64 \pm 0.32$ | $0.75 \pm 0.28$ |
| SAGE-TRSwap | 80 | $0.61 \pm 1.29$ | $\mathbf{123.0 \pm 65.6}$ | $\mathbf{1.2 \pm 0.5}$ | $\mathbf{42.4 \pm 14.1}$ | $\mathbf{0.53 \pm 0.18}$ | $\mathbf{0.99 \pm 0.03}$ |
| Ledidi-ST | 80 | $\mathbf{0.60 \pm 1.37}$ | $1181.6 \pm 386.0$ | $10.0 \pm 5.0$ | $50.6 \pm 17.4$ | $0.63 \pm 0.22$ | $0.80 \pm 0.21$ |

Table 1: Aggregate results across **12** regulatory models and **5** seeds (42–46) under edit budgets $B \in \{20, 40, 80\}$. Values are mean $\pm$ SD over TF×seed pairs. Clusters use a 20-bp gap threshold.

Attribution agreement also improves: edits made by SAGE-TRSwap more frequently fall on high-$|s|$ positions, increasing AP and rank-based alignment metrics. Qualitatively, SAGE-TRSwap tends to replace low-saliency "filler" edits with a smaller number of high-impact edits localized to one salient neighborhood. In the GATA2 case study (Fig. 2F–G), DeepSHAP logos within representative high-IG windows show that SAGE-TRSwap concentrates attribution around a compact motif-like pattern after editing, whereas Ledidi-ST exhibits a more diffuse attribution footprint.

## 6 LIMITATIONS

Attribution maps are imperfect: Integrated Gradients depends on the baseline and step count, and can be noisy or saturate, so a trust region derived from a single saliency map may miss distributed or redundant mechanisms. Our compactness metrics (span/clusters) are structural proxies for interpretability, not a guarantee of mechanistic validity, and highly localized edits may still be biologi-

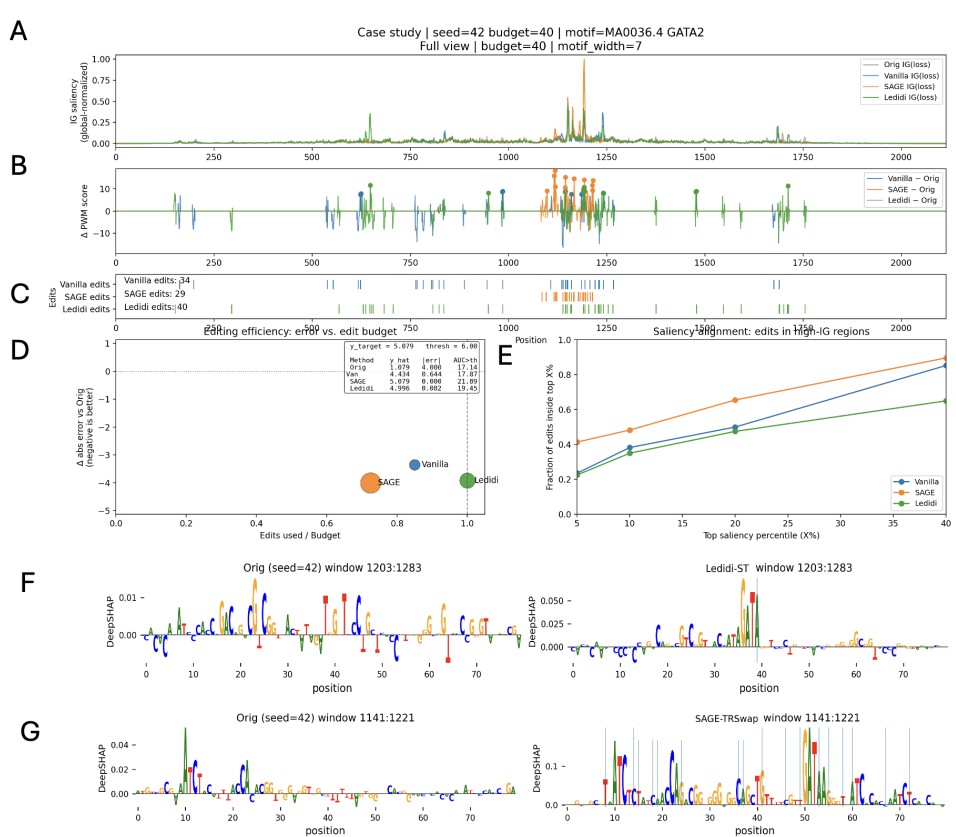

Figure 2: **Qualitative case study (GATA2; seed=42; budget** $B = 40$**; motif MA0036.4). (A)** Integrated Gradients (IG) on the squared error $(f(x) - y^\star)^2$ (per-position sum over bases; globally normalized across methods for this example). **(B)** Per-position changes in motif log-odds score relative to the original sequence (Method − Orig), taking the maximum over both strands; lollipops mark the top $K = 12$ positive increases per method. **(C)** Edited positions (rug plots; counts annotated). **(D)** Editing efficiency: $x =$ edits used / budget, $y = \Delta|f(x) - y^\star|$ vs. Orig (negative is improvement); marker area $\propto$ motif AUC-excess above threshold (thresh $= 6.0$). **(E)** Saliency alignment curve: fraction of edits falling in the top-$X\%$ of the *original* IG map for $X \in \{5, 10, 20, 40\}$. **(F–G)** DeepSHAP sequence logos for representative high-IG windows, comparing the original sequence (left) to the edited sequence from Ledidi-ST (F, right) and SAGE-TRSwap (G, right).

cally implausible (e.g., dense substitutions that disrupt chromatin context). Finally, our conclusions are based on model-based oracles and a fixed set of targets and budgets; while we use identical prediction losses for fair comparison, improved edit structure does not imply improved real-world regulatory function. Future work includes testing alternative attributions and stability criteria, learning adaptive trust-region widths that can accommodate multi-locus mechanisms, and validating designed edits with motif/grammar constraints and experimental assays.

## MEANINGFULNESS STATEMENT

Meaningful biological representations connect predictive signals to mechanistic, testable hypotheses. In sequence models, attribution maps provide a representation of positional importance, but design methods often ignore this representation and return scattered edit sets that are difficult to interpret. Our method uses attribution directly to constrain and refine the search, selecting solutions that both match target predictions and concentrate edits within salient regions. This yields

design proposals that are easier to rationalize (motif- or locus-level hypotheses) and more practical to validate, without retraining models or adding supervision.

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
