# OpenReview forum: "TRUST-REGION SALIENCY-GUIDED LOCAL SEARCH FOR INTERPRETABLE SEQUENCE DESIGN AT FIXED EDIT BUDGETS"
_ICLR.cc/2026/Workshop/LMRL — ICLR 2026 Workshop LMRL Poster_

### Official Review · Reviewer_T7oA · 2026-02-24
**In summary, this study substantially advances interpretable sequence design, supported by strong empirical results. I recommend this work as a valuable addition to the field**

**Rating:** 8
**Confidence:** 4

**Review:**

The article "TRUST-REGION SALIENCY-GUIDED LOCAL SEARCH FOR INTERPRETABLE SEQUENCE DESIGN AT FIXED EDIT BUDGETS" introduces SAGE-TRSwap, a saliency-guided local search for sequence design with fixed edit budgets to improve structural interpretability while preserving predictive performance. The main contribution is leveraging attribution maps to direct optimization, yielding more compact, interpretable edits. Results show reduced edit dispersion and more targeted high-saliency regions, with prediction error remaining steady or improving relative to baselines.
The work has clear motivation, a well-defined method, and consistent evaluation, and the approach is simple and works with existing models. Limitations include strong reliance on Integrated Gradients, lack of biological validation, and reliance on structural proxy interpretability metrics. There are also a few method comparisons, no theoretical analysis of the trust region, and a narrow experimental scope.
In summary, this study substantially advances interpretable sequence design, supported by strong empirical results. While additional robustness analyses, functional validation, and deeper theoretical justification could strengthen its impact, I recommend this work as a valuable addition to the field.

---

### Official Review · Reviewer_Pd1M · 2026-02-25
**Methodologically Sound Local Search Refinement with Limitations in Capturing Higher-Order and Biological Interactions**

**Rating:** 7
**Confidence:** 3

**Review:**

## Summary

The authors propose **SAGE-TRSwap**, a saliency-guided trust-region local search method for discrete sequence design under fixed Hamming edit budgets. The method optimizes the same predictive objective as a Ledidi-style relaxation baseline but biases edits toward high-attribution regions and introduces budget-preserving SWAP refinements. Across multiple regulatory targets and budgets, the approach substantially reduces mutation span and cluster count while maintaining or improving predictive error. The central goal is to improve interpretability by producing compact, attribution-aligned edit sets without sacrificing target matching.

---

## Pros

- The manuscript presents the problem statement clearly and is structured in an easily readable manner. The key innovations are directly highlighted and the methodological choices are well explained.

- The authors correctly identify a practical issue in existing relaxation-based approaches: highly dispersed edits can be difficult to interpret and experimentally validate. The focus on structural compactness and edit clustering directly addresses this limitation.

- The interpretability metrics (edit span, cluster count, saliency alignment) are clearly defined and consistently applied. These provide concrete operational proxies for structural interpretability.

- The empirical evaluation is relatively thorough, covering multiple regulatory targets, seeds, and budgets. Appropriate baselines are included to contextualize improvements.

- The SWAP refinement mechanism is simple but effective, and the use of attribution maps as trust regions is a lightweight yet principled modification to existing pipelines.

---

## Cons

- Despite the clear structure, parts of the manuscript are densely written and at times difficult to follow. The presentation occasionally feels compressed, likely due to page limits, but this affects readability.

- As identified by the authors, questions remain regarding the ability of saliency-guided trust regions to capture epistatic or multi-locus interactions. Many regulatory mechanisms are not purely first-order, and reliance on single-map attribution may bias the search toward locally salient positions while missing distributed cooperative effects. The real-world utility of this tool in modeling complex regulatory grammar therefore remains uncertain and would require further validation.

- While structural compactness is convincingly improved, compactness is only a proxy for interpretability. The manuscript does not provide mechanistic validation (e.g., motif enrichment statistics across targets, grammar-level analysis, or synthetic ground-truth interaction tasks) to demonstrate that the more localized edits correspond to biologically meaningful regulatory hypotheses rather than model-specific artifacts.

- A more granular case study examining specific mutations and proposing concrete biological hypotheses would have strengthened the interpretability claim. However, it is reasonable to assume that page limits constrained the depth of such analysis.

---

Overall, the method appears sound and well grounded in theory. Its use in sequence design as an interpretable alternative to existing paradigms makes it a worthwhile contribution for the workshop.

---

### Meta-Review · Area_Chair_zPH7 · 2026-02-27

**Recommendation:** Accept (Poster)
**Confidence:** 4

**Metareview:**

Accept.

---

### Decision · Program_Chairs · 2026-03-02

**Decision:**

Accept (Poster)

**Comment:**

Please see the meta-review.